# A steady-state model of microbial acclimation to substrate limitation

**John R. Casey** [1,2]*, **Michael J. Follows**[1]

**1** Department of Earth, Atmospheric and Planetary Sciences, Massachusetts Institute of Technology, Cambridge, Massachusetts, United States of America, **2** Department of Oceanography, University of Hawai'i at Mānoa, Honolulu, Hawai'i, United States of America

* jrcasey@mit.edu

## Abstract

Microbes acclimate to changes in substrate availability by altering the number of transporters on the cell surface, however there is some disagreement on just how. We revisit the physics of substrate uptake and consider the steady-state scenario whereby cells have acclimated to maximize fitness. Flux balance analysis of a stoichiometric model of *Escherichia coli* was used in conjunction with quantitative proteomics data and molecular modeling of membrane transporters to reconcile these opposing views. An emergent feature of the proposed model is a critical substrate concentration $S^*$, which delineates two rate limits. At concentrations above $S^*$, transporter abundance can be regulated to maintain uptake rates as demanded by maximal growth rates, whereas below $S^*$, uptake rates are strictly diffusion limited. In certain scenarios, the proposed model can take on a qualitatively different shape from the familiar hyperbolic kinetics curves, instead resembling the long-forgotten Blackman kinetics.

**Data Availability Statement:** All relevant data are within the paper and its Supporting Information files. Accompanying code and documentation is available at https://github.com/jrcasey/NutrientUptake.

## Author summary

The mechanics of resource-limited microbial growth is a fundamental focus in cell biology and biophysics. Physiological acclimation plays a key role in microbial growth rate dependence on the availability of a limiting resource, but progress has been mostly rooted in theoretical studies due to a lack of relevant experimental data. In light of new quantitative proteomics data which disagree with current models, we revisited the physics of substrate transport and propose a model, based on a different set of assumptions, which applies to the steady-state scenario. Depending on the design of the transport system, the proposed model predicts that microbial growth rate dependence on substrate availability can take on a familiar hyperbolic shape (e.g., Monod) or a piecewise linear shape, not unlike the Blackman kinetics model which fell out of favor long ago. The sharp transition appears from a discontinuity which marks a critical substrate concentration, above which physiological acclimation can sustain maximal growth rates, and below which diffusion is strictly limiting. We implemented the model in *Escherichia coli* using a combination of flux balance analysis, molecular modeling, and quantitative proteomics data. Predicted kinetics

**Funding:** This work was supported by the Simons Foundation (https://www.simonsfoundation.org) as part of the Simons Collaboration on Computational Biogeochemical Modeling of Marine Ecosystems (https://cbiomes.org; Simons Foundation grant no. 549931 to M.J.F. and grant no. 549894 to J.R.C). The funders had no role in study design, data collection and analysis, decision to publish, or preparation of the manuscript.

**Competing interests:** The authors have declared that no competing interests exist.

closely matched experimental observations across a range of carbon substrates, and the model can be easily implemented for other systems.

## Introduction

Microbial growth rates may be limited by catalytic rates or by the rate of diffusion of resources, which are often supplied with some irregularity in the natural environment. As the availability of nutrients and organic substrates change, various metabolic and physiological acclimation strategies can be leveraged to maximize growth rate and remain competitive. For instance, flexibility in the molecular composition of biomass can be used to compensate for resource deficits by adjusting demands through holistic changes in elemental stoichiometry. Given the dynamic nature of microbial ecosystems, these acclimated phenotypes are probably the norm rather than the exception.

An important physiological acclimation strategy to maximize growth on a limiting resource is to alter the number of transporters on the cell surface; however, two conflicting views of how microbes implement this strategy have emerged (Fig 1). One view ("Model A") predicts that, at very low substrate concentrations, the diffusive flux is low and therefore very few transporters would be required to match uptake rates to encounter rates. As substrate concentrations increase, uptake rates become limited by the number of transporters, so more are synthesized [1–8]. Another view ("Model B") predicts the opposite; at low substrate concentrations the encounter rate is limiting, so high transporter abundance is favored, while at higher concentrations resources are allocated for use elsewhere in the proteome [9–13]. Considering the evolution of these two divergent views, it is interesting that both evolved under different assumptions from the same underlying physics [14].

The relevant data (i.e., the abundance of transporters across a range of substrate concentrations) which could resolve, or even reconcile these opposing views has, until relatively recently, been lacking; however, progress in proteomics has enabled the direct quantitation of the majority of expressed proteins in model organisms like *Escherichia coli* K12 [15], including its transporters. In glucose fed batch cultures and chemostats across a range of dilution rates,

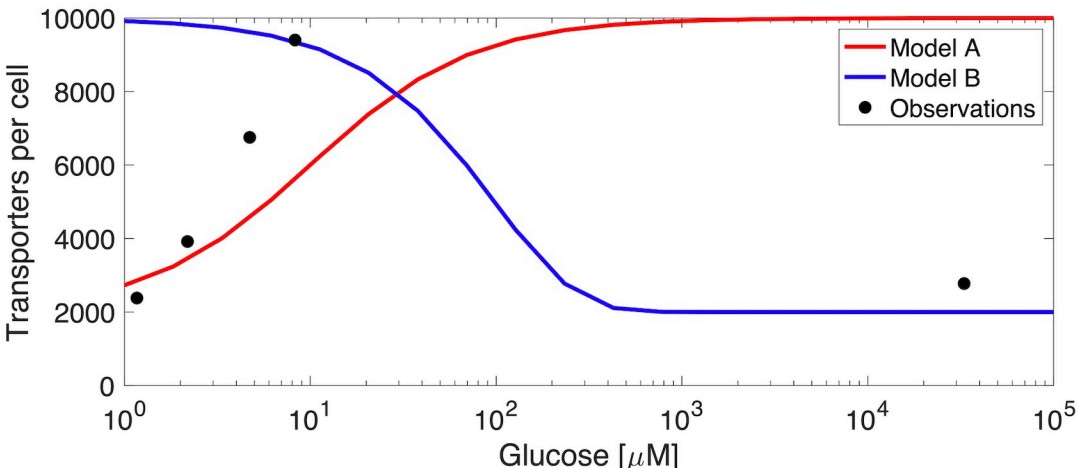

**Fig 1. Observed glucose transporter abundances across a range of glucose concentrations ([15]; data plotted with permission from the authors).** Qualitative predictions of Models A and B are shown to illustrate their expected behavior.

transporter abundances qualitatively matched neither of the two previous model predictions (Fig 1). This finding suggests perhaps that another interpretation of the underlying physics be posed which explains these observations. For instance, prior approaches in line with Model B have leaned heavily on resource allocation, whereby a tradeoff is established by the fraction of the proteome dedicated to substrate transport [8–12]. This fraction was assumed to modulate both the affinity and the maximum uptake rate, often requiring large proteome investments in transport (e.g., up to 77% of total protein; [8]), however the total contribution of all 90 transporter proteins on both the inner and outer membranes and their associated periplasmic binding proteins to the *Escherichia coli* K12 proteome was 11±3% across 22 different growth conditions, with any particular transporter complex contributing much less [15]. Clearly the allocation of resources to transport is not without cost, as we will explore, but may offer confounding predictions when modeled as a linear constraint on kinetic parameters.

We consider the simplest case of a single cell growing on a single limiting substrate $S$ (Fig 2). The rate of change of the cell quota $Q$ of element $X$ (mol $X$ cell$^{-1}$) can be expressed in terms of $S$ as

$$\frac{dQ}{dt} = v(r, n, S, D)Y_{X/S} - \mu Q - MQ, \tag{1}$$

where $v$ is the substrate uptake rate (mol $S$ cell$^{-1}$ s$^{-1}$), $r$ is the cell radius (m), $n$ is the number of transporters (cell$^{-1}$), $D$ is the molecular diffusivity of the substrate (m$^2$ s$^{-1}$), $Y_{X/S}$ is the yield (mol $X$ [mol $S$]$^{-1}$), and $M$ is a non-growth associated maintenance requirement (s$^{-1}$). Over the time-average in a continuous culture, $\frac{dQ}{dt} = 0$, so we define the steady-state growth rate $\mu$ (s$^{-1}$)

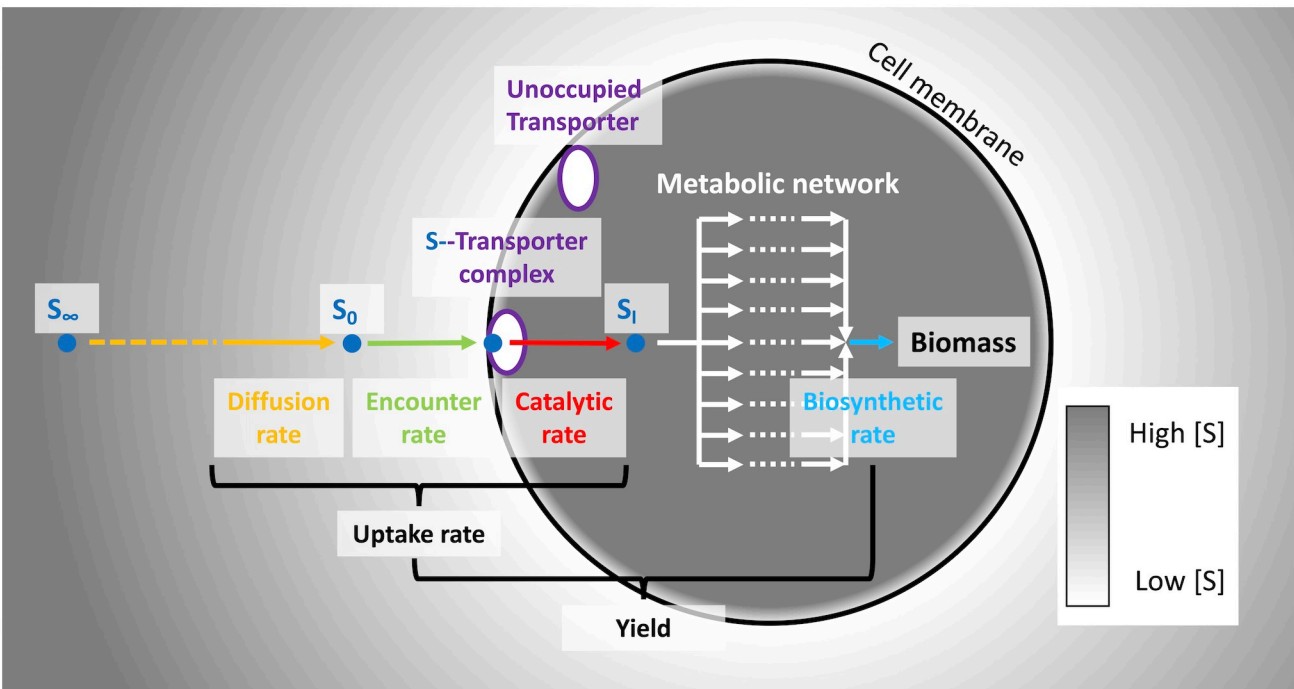

**Fig 2. Conceptual diagram of substrate uptake.** A substrate molecule diffuses down a gradient in concentration from $S_\infty$ toward the cell surface $S_0$. The molecule encounters an unoccupied transporter at a rate no higher than the catalytic rate, where it is subsequently transported to the interior $S_I$ and metabolized to generate reducing power or to synthesize new biomass. Coupling of a mechanistic model of substrate transport with flux balance analysis of a stoichiometric model of the metabolic network allows computation of yields, optimal uptake rates, and optimal transporter abundances.

as

$$\mu = \frac{v(r, n, S, D)}{Q} Y_{X/S} - M. \tag{2}$$

The experimental data of Shmidt et al., [15] provide the opportunity to test and calibrate a model of nutrient uptake and transporter allocation since $r$, $n$, $S$, $D$, and $\mu$ are simultaneously known. Here we base the functional form of $v$ on the model of Armstrong [4], an approximation to the model of Pasciak and Gavis [2] (described in detail in the Model section). To close the model according to Eq (2) we must also constrain the yield $Y_{X/S}$ and maintenance $M$; we do so by employing a genome-scale metabolic model of *Escherichia coli* K12 [16].

In short, we will demonstrate that the model is compatible with the quantitative proteomic data of Schmidt et al., [15] under the assumption that allocation to transporters is optimized to two, theoretically anticipated regimes: (i) a regime where internal synthesis rates are the limiting factor and (ii) a regime where diffusive encounter with the resource is the limiting factor.

## Results

### Conceptual model

For simplicity, we consider a single cell of radius $r$ suspended in an infinitely large volume, containing a single nutrient at bulk concentration $S_\infty$ (mol m$^{-3}$; Fig 2; Table 1). The subscript $\infty$ denotes that the concentration is that of a volume an infinite distance from the cell surface. A crude description of transport of an extracellular substrate at a concentration at the surface of the cell $S_0$ (mol m$^{-3}$) across the inner membrane via a transporter $E$ to the intracellular space at a concentration $S_i$ can be expressed as the two-step reaction

$$S_0 + E \xrightarrow{k_1} ES \xrightarrow{k_{cat}} E + S_i, \tag{3}$$

where $k_1$ is the molecular encounter rate with a transporter (mol m$^{-3}$ s$^{-1}$) and $k_{cat}$ is the catalytic constant or turnover number, the rate of dissociation of the transporter-substrate complex $ES$ in the transport process (mol transporter$^{-1}$ s$^{-1}$). It should be noted that $k_1$ represents not only the encounter rate, but specifically the encounter rate of molecules colliding with the transporter with sufficient energy to overcome a reaction energy barrier. Thus, the back reaction, often denoted $k_{-1}$, is implicit. Also note that $k_{cat}$ should be interpreted to be the maximum apparent catalytic rate *in vivo*, rather than the constitutive *in vitro* $k_{cat}$ [17].

At high substrate concentrations ($S_0 \to \infty$) when the molecular encounter rate with each transporter far exceeds their maximum catalytic rate ($k_1 \gg k_{cat}$), effectively all transporters are occupied, that is, in the process of carrying out a transport reaction. For consistency with previous studies, we refer to this limit as the porter limit, denoted by the superscript $P$. At the porter limit, the maximum uptake rate $v^P_{max}$ is set by the catalytic rate of a number of transporters $n$,

$$v^P_{max} = nk_{cat} \tag{4}$$

However, another limit is reached as transporter abundances increase further; we refer to this limit condition as the growth limit $v^G_{max}$, denoted here and elsewhere by the superscript $G$. In this state, increased uptake capacity would exceed metabolic demands and thus the excess transporters would be proportionately inhibited, so as not to violate $\frac{dQ}{dt} = 0$. The maximum growth rate under optimal conditions can be set by a variety of internal bottlenecks (*e.g.*, protein translation rates, oxygen diffusion, metabolic choke-points, molecular crowding,

**Table 1. Parameters used in this article.**

| Parameter | Description | Units |
|---|---|---|
| $\alpha$ | Substrate molecule capture probability | dimensionless |
| $\mu$ | Growth rate | $s^{-1}$ |
| $v^D$ | Diffusive molecular encounter velocity | $m\ s^{-1}$ |
| $v^P$ | Membrane transport velocity at the porter limit | $m\ s^{-1}$ |
| $\phi$ | Shape coefficient | dimensionless |
| $A$ | Transporter capture area | $m^2$ |
| $D$ | Hydrated molecular diffusivity | $m^2\ s^{-1}$ |
| $f_{max}$ | Maximum fraction of cell surface area for transporters | dimensionless |
| $k_s$ | Half-saturation concentration | $mol\ m^{-3}$ |
| $k_s^P$ | Half-saturation concentration at the porter limit | $mol\ m^{-3}$ |
| $k_s^D$ | Half-saturation concentration at the diffusive limit | $mol\ m^{-3}$ |
| $k_{cat}$ | Catalytic rate of an individual transporter | $mol\ s^{-1}$ |
| $l^R$ | Reaction-diffusion characteristic length scale | m |
| $n$ | Number of transporters | $cell^{-1}$ |
| $n_{max}$ | Maximum number of transporters | $cell^{-1}$ |
| $n_G$ | Number of transporters of replete batch-acclimated cells | $cell^{-1}$ |
| $n^*$ | Optimal number of transporters for any phenotype | $cell^{-1}$ |
| $n_G^*$ | Optimal number of transporters at the growth limit | $cell^{-1}$ |
| $n_D^*$ | Optimal number of transporters at the diffusion limit | $cell^{-1}$ |
| $Q$ | Cellular biomass quota | $mol\ X\ cell^{-1}$ |
| $r$ | Cell radius | m |
| $S_\infty$ | Ambient substrate concentration | $mol\ m^{-3}$ |
| $S^*$ | Nutrient limitation concentration | $mol\ m^{-3}$ |
| $S_G^{lb}$ | Lower bound of the $n_G^*$ feasible domain | $mol\ m^{-3}$ |
| $S_{SA}^{lb}$ | Lower bound of surface area limitation | $mol\ m^{-3}$ |
| $S_{SA}^{ub}$ | Upper bound of surface area limitation | $mol\ m^{-3}$ |
| $Sh$ | Sherwood number | dimensionless |
| $v$ | Uptake rate | $mol\ s^{-1}$ |
| $v_{max}$ | Maximum uptake rate | $mol\ s^{-1}$ |
| $v_{max}^G$ | Maximum uptake rate of replete batch-acclimated cells | $mol\ s^{-1}$ |
| $Y_{X/S}$ | Biomass yield | $mol\ X\ [mol\ S]^{-1}$ |

temperature), which are manifested by differences in metabolic and physiological designs in a particular environment. The growth limit represents a specific instance of $v_{max}$ where cells have acclimated to growth at high nutrient concentrations by adjusting the number of transporters to some optimal abundance $n^* = n^G$ which supplies substrate at the rate required to satisfy the maximum growth rate. In this instance, the porter limit and growth limit converge.

## Kinetic model

A model of the dependence of uptake rate kinetics on changes in cell physiology was previously developed [2]. In subsequent work, Armstrong [4] provided a convenient approximation of Pasciak and Gavis' quadratic model, resembling an expansion of the hyperbolic Michaelis-Menten model $v = \frac{v_{max}S_\infty}{k_s + S_\infty}$. In this approximation, the apparent half-saturation concentration $k_s = k_{cat}/k_1$ can be described by the sum of two limits: a porter limit $k_s^P$ and a diffusion limit $k_s^D$. Armstrong's model incorporates similar dynamics to the quadratic model, but uses the

simplifying assumption that cell radii are much smaller than, or much larger than a characteristic length scale of the reaction-diffusion process $l^R = v^P_{max}/k^P_s \phi Sh D$. A discussion on the validity of this assumption can be found elsewhere [6]. Thus, we apply Armstrong's approximation

$$v = \frac{n k_{cat} S_\infty}{k^P_s + k^D_s + S_\infty},$$ (5)

where $k^P_s$ is the effective half-saturation concentration at the porter limit, which is independent of the number of transporters,

$$k^P_s = \sqrt{\frac{A}{\pi}} \frac{\pi k_{cat}}{4\alpha A D \sqrt{A\pi}}.$$ (6)

Critically, $\alpha$ represents a dimensionless probability that a substrate molecule which enters the vicinity of the transporter catchment area $A$ ($m^2$) is captured and transported. This parameter indirectly accounts for the fraction of collisions which exceed the activation energy of the ligand binding reaction, without any explicit knowledge of the magnitude of this barrier or the energy of the collision,

$$\alpha = \frac{v \sqrt{A\pi}}{4D}.$$ (7)

We derive this probability $\alpha$ by equating the velocity (m s$^{-1}$) of molecular transport across the membrane when all transporters are saturated ($v^P$ at the porter limit; [4]) to the velocity of the nutrient molecule diffusing towards the cell surface ($v^D$; [14]), which is dependent on the hydrated molecular diffusivity of the substrate $D$ ($m^2$ s$^{-1}$), the cell size, and the advective velocity of the cell $u$ (m s$^{-1}$),

$$v^D = \frac{D}{r} + \frac{u}{2}.$$ (8)

Thus, motility is inversely proportional to the effective half-saturation concentration at the porter limit, by increasing the encounter rate $k_1$ relative to the catalytic rate, as can be seen from Eqs (6), (7) and (8). The effective half-saturation concentration at the diffusion limit $k^D_s$ is not only cell size-dependent, but is dependent on the number of transporters, such that

$$k^D_s = \frac{n k_{cat}}{\phi Sh D r_0},$$ (9)

where $\phi$ is a dimensionless cell shape factor and $Sh$ is the dimensionless Sherwood number, which relates mass transfer by advective shear forces to those of the viscous forces. We include $\phi$ and $Sh$ for completeness, but have excluded any explicit treatment of their effects on nutrient transport in this article by assigning both of their values to be 1. Both turbulence and cell shape influence the encounter rate, and a detailed discussion on their effects on nutrient transport can be found elsewhere [18, 19].

## Acclimation

Afforded sufficient time, microbial isolates grown under viable conditions will acclimate both metabolically and physiologically so as to maximize growth rate (e.g., [20]). Given this observation, it follows that uptake rates should be matched as closely as possible to $v^G_{max}$ by regulating transporter abundances optimally. Maintenance of a number of transporters in excess of this

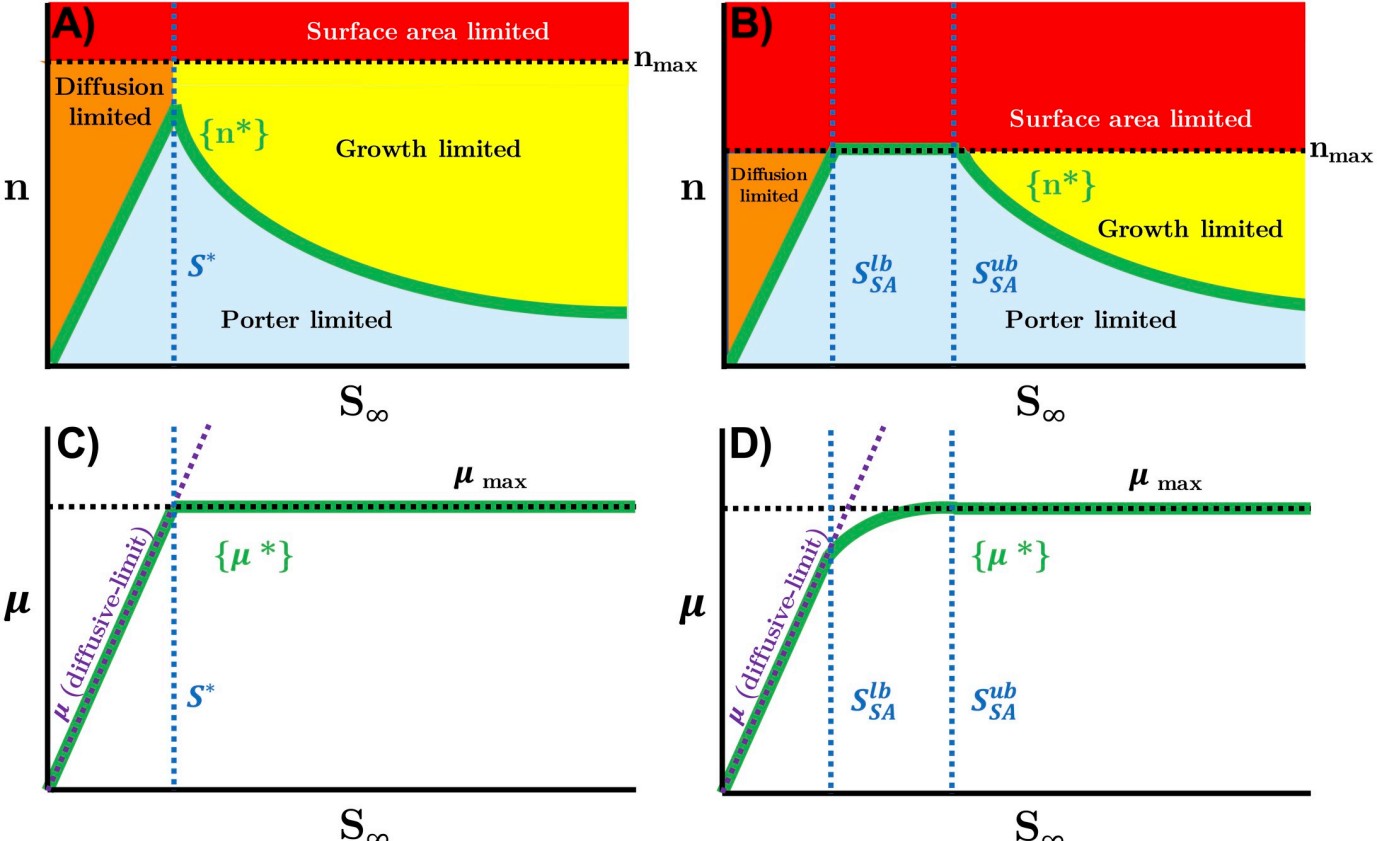

**Fig 3. Schematic of steady-state nutrient acclimation.** A)—Optimal transporter abundances $n^*$ (solid green line) lie at the porter limitation (blue shaded region) boundary. All points above and below this boundary fail to maximize growth rate; for all points below, the uptake rate is sub-maximal; for all points above, some transporters are unoccupied. Over an interval of bulk substrate concentrations $S_\infty \leq S^*$, this boundary is met with diffusion limitation (orange shaded region), where the catalytic rate exceeds the encounter rate; as concentrations exceed $S^*$, the porter limitation boundary transitions to an internal growth rate limit (yellow shaded region), where the catalytic rate exceeds the rate of some downstream reaction. B)—For smaller cells, or for transporters with slower $k_{cat}$, membrane surface area limitation (a special case of porter limitation) may be encountered. In the interval bounded by $S^{lb}_{SA}$ and $S^{ub}_{SA}$, $n^*$ is constrained to $n_{max}$ (black dashed line). C)— Growth rates of optimally acclimated cells (solid green line) follow the intersection of two limits: the diffusive limit (purple dashed line) and the internal growth limit $\mu^G_{max}$ (black dashed line). These rate limits are, again, bisected by $S^*$ with a sharp transition. D)—If surface area limitation is encountered, growth rates in the interval $S^{lb}_{SA} \leq S_\infty \leq S^{ub}_{SA}$ follow a more gradual, hyperbolic transition from diffusion limitation to growth limitation.

optimal value would result in the intracellular accumulation of the substrate and allosteric feedback inhibition of metabolism and sub-optimal growth. Conversely, maintenance of an insufficient number of transporters would result in porter limitation, which is also a sub-optimal growth state. If both of these statements are true, then an optimal number of transporters $n^*$ should be predicted over two concentration intervals; a zero-order interval over which uptake rates may be maintained at $v^G_{max}$, and a first-order interval over which uptake is limited by the substrate concentration dependent diffusive flux. A schematic of these transitions is given in panels A and C in Fig 3. At zero-order,

$$v = v^G_{max} = n^G k_{cat} = \frac{n^* k_{cat} S_\infty}{k^P_s + \frac{n^* k_{cat}}{\phi ShDr} + S_\infty}. \tag{10}$$

By demanding that the uptake rate match $v_{max}^G$, we find the optimal number of transporters to be

$$n_G^* = \frac{k_s^P + S_\infty}{\frac{S_\infty}{n^G} - \frac{k_{cat}}{\phi ShDr}}.$$  (11)

In this expression, $n_G^*$ behaves proportionally to $1/S_\infty$ and it is plainly seen that a discontinuity is possible at a particular $S_\infty$. This concentration $S_G^{lb}$ represents the lower limit for which Eq 11 holds,

$$S_G^{lb} = \frac{n^G k_{cat}}{\phi ShDr} = \frac{v_{max}^G}{\phi ShDr}.$$  (12)

For all concentrations $S_\infty \leq S_G^{lb}$, no abundance of transporters is sufficient to match uptake to the growth limit. Instead, the maximal uptake rate is set by the diffusive flux $v^D = 4\pi\phi ShrDS_\infty$. Accordingly, the optimal number of transporters $n_D^*$ in this diffusion limited interval is

$$n_D^* = \frac{v^D}{k_{cat}}.$$  (13)

In keeping with the maximum growth rate assumption, it follows that maintaining unoccupied transporters incurs some non-zero cost; we therefore consider the minimum $n^*$ over the full domain of $S_\infty$. The intersection of the two transporter abundance optima $S^*$ represents the steady-state transition from diffusion limitation to growth limitation. Since $n_D^*$ is a linear function of $S_\infty$, and since $n_G^*$ is symmetric about the discontinuity $S_G^{lb}$, the intersection $S^*$ (in the positive domain of $S_\infty$) is the positive root of a quadratic,

$$0 = \frac{rDS_\infty^2}{k_{cat}} - 2S_\infty - k_s^P.$$  (14)

By assuming optimal transporter abundance within the cellular growth context of Eq 2, we arrive at the proposed steady-state acclimation model (Fig 3), which is piecewise linear with a transition at the discontinuity $S^*$,

$$\mu = \begin{cases} \frac{4\pi rDS_\infty}{Q} Y_{X/S} - M & \text{if } S_\infty \leq S^* \\ \\ \frac{v_{max}^G}{Q} Y_{X/S} - M & \text{if } S_\infty > S^* \end{cases}$$  (15)

## Cell size and surface area limitation

It is worth noting that although cell size is clearly an important determinant of growth rate in the diffusion limited regime, the range of cell sizes observed within the range of phenotypes from batch-acclimated to the lowest dilution rate glucose-limited chemostat spanned only a 12% change in radius [15]. Thus we expect acclimation to nutrient availability in this case to be predominantly regulated by transporter abundance. It might also be noted that despite this small observed change in cell sizes over the substrate concentration interval tested, the most effective strategy to maximizing growth rate as $S_\infty \to 0$ shifts from regulating $n$ to regulating $r$.

Perhaps *Escherichia coli* K12 is simply not capable of this adaptation, but should not be ignored in the case of other taxa which are.

As concentrations approach $S_G^{lb}$ in the nutrient-replete regime, $n^* \to \infty$, but clearly this is not physiologically possible since there is finite membrane space afforded to transporters. In their conceptual model of transport through a sphere of known dimension and with a number $n$ of pores of known dimension, Berg and Purcell [1] proposed that the addition of a number of transporters $n \geq \frac{\pi r}{s}$, where $s$ is the radius of the transporter, does not appreciably increase the uptake rate [1]. Given their selection of 10 Å for transporter radius and for a cell of 1 $\mu$m radius, this corresponds to an areal coverage of less than 0.1%. Although this is a convenient parameterization for the maximum number of transporters $n_{max}$, we find that in the case of the glucose transporter PtsG, given experimental maximum $n$ = 9402 cell$^{-1}$ [15] and corresponding cell size, $s$ would need to be no larger than 2 Å which is roughly half the van der Waals radius of the glucose molecule alone, and much smaller than the barrel radius from our structural analysis (see Methods). On the contrary, Asknes and Egge [3] predicted a site coverage of 8.5%, corresponding to $n_{max}$ = 21, 058 cell$^{-1}$, which is comfortably above the maximum observed $n$ given our best estimates of the corresponding catchment area for the PtsG protein. Therefore, we required that all $n \leq n_{max}$, where $n_{max} = \frac{4\pi r^2 f_{max}}{A}$ and the maximal areal coverage $f_{max}$ = 0.085. Nevertheless, the wide discrepancy between $f_{max}$ estimates between these studies is unclear, and it is certainly possible that $f_{max}$ will vary between cell membrane designs, and is therefore a somewhat unsatisfying and arbitrary boundary which should be better constrained.

Variation in cell size introduces a nuance to our proposed model which provides a mechanistic basis for more gradual transitions observed between the diffusion-limited regime and the growth-limited regime. In some scenarios, $n^*$ exceeds $n_{max}$ over a substrate concentration interval wherein uptake rates become surface area (SA) limited, a special case of porter limitation. This interval can be constrained to a lower $S_{SA}^{lb}$ and upper bound $S_{SA}^{ub}$, both evaluated at the intersections of Eqs 11 and 13 with $n_{max}$,

$$S_{SA}^{lb} = \frac{n_{max} k_{cat}}{4\pi r D} \tag{16}$$

$$S_{SA}^{ub} = \frac{\frac{n_{max} k_{cat}}{\phi S h D r} + k_S^P}{\frac{n_{max}}{n^G} - 1} \tag{17}$$

Within the interval $S_{SA}^{lb} \leq S_\infty \leq S_{SA}^{ub}$, the optimal number of transporters is constrained to $n_{max}$, resulting in a hyperbolic transition in uptake rate from diffusion limitation to growth limitation (panels B and D in Fig 3). Although it was not relevant for our present case, to accommodate surface area limitation under such a scenario, a third conditional is appended to Eq 15,

$$\mu = \begin{cases} \dfrac{4\pi r D S_\infty}{Q} Y_{X/S} - M & \text{if } S_\infty < S_{SA}^{lb} \\[2ex] \dfrac{v(n_{max})}{Q} Y_{X/S} - M & \text{if } S_{SA}^{lb} \leq S_\infty \leq S_{SA}^{ub} \\[2ex] \dfrac{v_{max}^G}{Q} Y_{X/S} - M & \text{if } S_\infty > S_{SA}^{ub} \end{cases} \tag{18}$$

where $v(n_{max})$ is the uptake rate defined in Eq 5, evaluated at $n = n_{max}$.

## Model validation

Glucose transport in aerobic cultures of the model bacterium *Escherichia coli* K12 was used as a system to validate our acclimation model. Although the physiological response to glucose limitation in *E. coli* is complex, with several complementary transporters with broad substrate specificity induced by cAMP, the primary transporter under both glucose excess and glucose limited growth conditions is the phosphotransferase system (PTS) which relies on the glucose-specific permease PtsG [27]. Indeed, PtsG was the most abundant permease in both glucose-limited chemostats and glucose excess batch cultures (S1 Fig; [15]).

Whereas $k_{cat}$ values are rarely reported for transporters, we were able to directly compare $k_s$ values from a collection of 11 substrates (Table 2). The modeled effective half-saturation concentrations ($k_s = k_s^P + k_s^D$) for all substrates spanned 4 orders of magnitude between substrates. $k_s$ values from experimental studies were compiled from the same *Escherichia coli* strain K12 (but not necessarily from the same sub-strain), harvested in exponential phase from substrate replete batch cultures. In a direct comparison, our model predictions agreed closely with these reported values (Fig 4; Model 2 linear regression, $R^2 = 0.89$, dF = 11), and the slope was not different from parity (regression slope = 0.98, standard error = 0.11). Note that the literature value reported for zinc was determined by a method which gives a value closer to the dissociation constant rather than the half-saturation constant [28], and should therefore be considered an upper bound. The deviation between predicted and observed $k_s$ values for acetate transport is less obvious, but may be due to either the mechanism of the transporter ActP, which is the only symporter in our set, or the presence of a second, lower-affinity symporter YaaH which would introduce biphasic kinetics that our model does not currently resolve.

Our core assumption, that unoccupied transporters would not be maintained by acclimated phenotypes which have optimized their physiology to maximize growth rates, can be tested by comparing transporter abundances for non-limiting substrates in cells grown across a range of growth conditions, with the expectation that abundances scale with maximal uptake rates. Additionally, at constant temperature, pH, and salinity, the property $k_{cat}$ would be expected to be invariant. Both the absolute abundance (cell$^{-1}$) and the aerial density (m$^{-2}$ of inner membrane surface area) of transporters of the inorganic nutrient ions phosphate, sulfate, and zinc, which were supplied in great excess in all media formulations [15], were linearly related to their $v_{max}^G$ uptake rates, as quantified by FBA, across all growth conditions ($P = 9 * 10^{-5}$, $5 *$

**Table 2. Summary of parameter values corresponding to batch growth conditions.** Sulfate, phosphate, and zinc parameters are reported only for glucose-replete batch cultures. "Exp." refers to experimentally determined $k_s$ values, except in the case of zinc, which should be considered a dissociation concentration and is thus an upper bound.

| Substrate | Gene | $A$ $10^{-16}$ m$^2$ | $v_{max}$ fmol cell$^{-1}$ h$^{-1}$ | $n^G$ cell$^{-1}$ | $k_{cat}$ molecules s$^{-1}$ | Model $k_s$ $\mu$M | Exp. $k_s$ $\mu$M | Reference |
|---|---|---|---|---|---|---|---|---|
| Sulfate | cysW | 1.62 | 0.14 | 39 | 595 | 3.8 | 2.1 | [29] |
| Phosphate | pstB | 0.87 | 0.53 | 113 | 786 | 9.3 | 5.5 | [30] |
| Zinc | znuA | 0.51 | 187 ($10^{-6}$) | 434 | 0.1 | 1.2 ($10^{-3}$) | 20 ($10^{-3}$) | [28] |
| Acetate | actP | 1.15 | 10.63 | 336 | 5293 | 34.8 | 5.4 | [31] |
| Fructose | fruA | 1.22 | 7.42 | 3364 | 369 | 6.9 | 5.4 | [32] |
| Fumarate | dctA | 1.42 | 8.72 | 596 | 2446 | 21.3 | 30 | [33] |
| Galactose | mglA | 0.15 | 2.33 | 534 | 732 | 66 | 59 | [34] |
| Glucose | ptsG | 0.39 | 6.37 | 2775 | 384 | 16.6 | 20 | [35] |
| Glycerol | glpF | 0.42 | 8.36 | 1289 | 1085 | 27.3 | 19 | [36] |
| Succinate | dctA | 1.42 | 8.17 | 567 | 2410 | 21.2 | 25 | [33] |
| Xylose | xylH | 1.61 | 6.99 | 105 | 11196 | 94.3 | 100 | [37] |

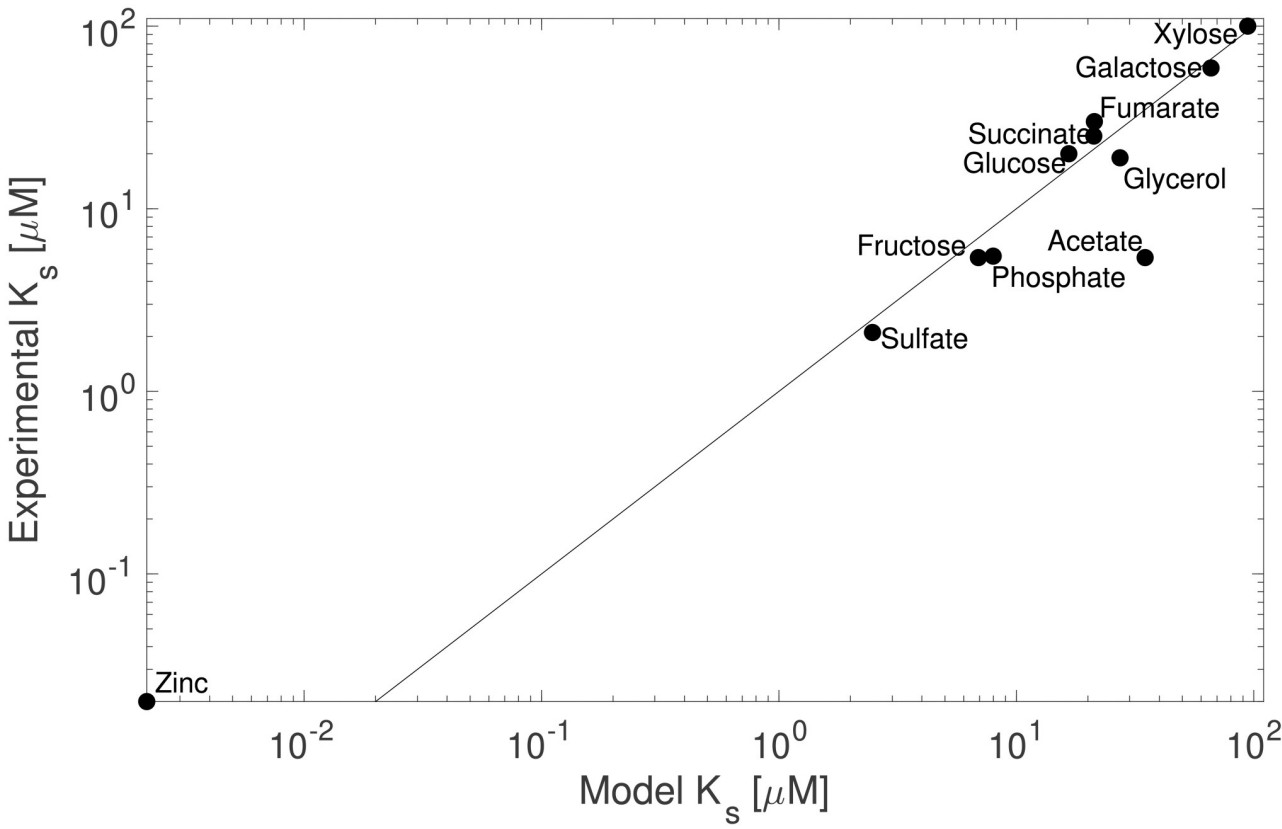

**Fig 4. Comparison of predicted and experimental values of the effective half-saturation concentrations of 11 substrates in nutrient-replete batch acclimated cultures of *Escherichia coli* K12.** Parity is indicated as a solid line.

$10^{-3}$, and $6 * 10^{-3}$, respectively). The coefficient of variation between growth conditions for $k_{cat}$ was less than 25% for this subset, and no growth rate or cell size dependence of the residuals could be detected.

## Acclimation to limiting nutrients

*Escherichia coli* K12 BW25113 cells maintained an optimal number of PtsG transporters corresponding to Eq 11 for all $S_\infty > S^*$, and corresponding to Eq (13) for all $S_\infty \leq S^*$. Fig 5 shows contours of uptake rates in the plane of transporter abundance and glucose concentrations. The instantaneous uptake rate kinetics for an arbitrary acclimated phenotype may be described by the hyperbolic curve generated by following a horizontal section in the top panel, corresponding to a constant number of transporters. Accordingly, the uptake rate kinetics for all optimally acclimated phenotypes may be described by following $n^*$, which is a linear function proportional to $S_\infty$ in the diffusion-limited regime, and a reciprocal function of $S_\infty$ in the growth limited regime. Predicted $n^*$ optima were in close agreement with measured PtsG glucose transporter abundances for *Escherichia coli* cultures grown in glucose-limited chemostats spanning a range of dilution rates from 0.12 h$^{-1}$ to 0.50 h$^{-1}$ and in nutrient replete batch acclimated cultures growing in exponential phase at 0.60 h$^{-1}$. In this particular scenario, $n_{max}$ was not intersected by $n^*$, so no intermediate surface area limitation transition was encountered (Fig 5—Top panel; [15]). Uptake rates for predicted optimally acclimated phenotypes also closely matched those derived from data (Fig 5—Bottom panel; [15]). The uptake rate kinetics

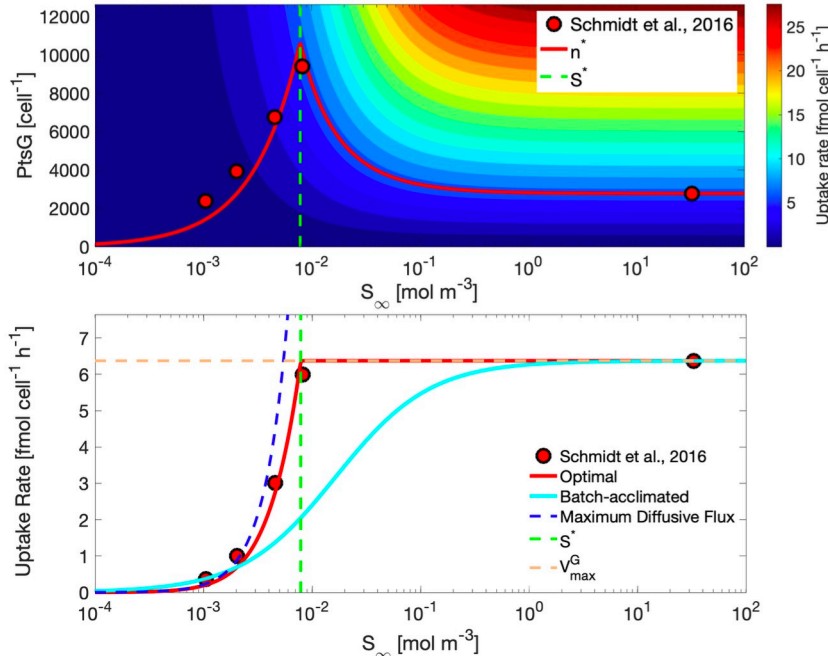

**Fig 5. Model predictions and observations of *Escherichia coli* acclimation to growth on glucose.** Top panel—Modeled and experimental abundances of the glucose transporter PtsG across steady-state concentrations of glucose (log scale). Contours indicate the corresponding uptake rates. The vertical green dashed line represents the critical substrate concentration $S^*$. $n_{max}$ is above the plotted range. *Bottom panel*—Modeled and experimental uptake rates over an interval of glucose concentrations. The uptake rate profile for a glucose-replete batch acclimated culture is shown in cyan. Uptake rates for all acclimated phenotypes over the concentration range are shown in red, with model predictions shown as a continuous line and experimentally derived values from published data [15] shown as markers. For guidance, the maximum diffusive flux is shown as a dashed blue line, $S^*$ is shown as a vertical dashed green line, and $v_{max}^G$ is indicated by the horizontal dashed orange line.

for all optimally acclimated phenotypes was linear in the diffusion limited regime, and transitioned at $S^*$ to maintain $v_{max}^G$ in the growth limited regime.

## Discussion

A model of substrate uptake which accounts for optimal physiological acclimation to sustained growth at a range of constant substrate concentrations was developed and implemented with a pseudo-mechanistic approach. Critically, the proposed model assumes that an optimal phenotype would maintain only the minimal number of transporters required to sustain either the maximum growth rate or the maximum diffusive flux. Maintaining more than this optimal abundance would incur a cost by exceeding substrate demand, and thus a growth rate penalty; maintaining fewer than this optimal abundance would result in sub-maximal substrate uptake rates, also a growth rate penalty. The resulting profile of both observed and optimal transporter abundances (Fig 5) shows an intermediate maximum, which is unexpected by conventional wisdom of, at least, the transcriptional response to nutrient limitation. The gap in observations of transporter abundances between the highest dilution rate glucose-limited chemostat and the glucose-replete batch acclimated culture is expected for practical reasons, but is somewhat unsatisfying and should be revisited to further validate the proposed model as quantitative proteomics datasets become more commonplace. Nevertheless, the steady-state kinetics associated with this optimal set of phenotypes (Eq 15) resembles those of the piecewise linear Blackman kinetics, and for similar reasons; a transition from one limitation to another. The critical

substrate concentration $S^*$ therefore represents the transition between diffusion limitation and growth limitation, barring the scenario where membrane surface area limitation mediates that transition. Variations in cell size or $k_{cat}$ introduce some nuance to the shape of this transition, as membrane surface area limitation presents another, intermediate limiting state. In those scenarios, $n^*$ is held at $n_{max}$, resulting in hyperbolic uptake rates in the interval between $S_{SA}^{lb}$ and $S_{SA}^{ub}$, thus providing a mechanism to reconcile observations of a more gradual transition from first-order diffusion limitation to zero-order growth limitation (Eq 18). Although we have considered the simple case of a single limiting substrate, we speculate that the competition between transporters for membrane space may give rise to apparent co-limitation when multiple substrates are present at concentrations near $S^*$, as may very well be the case for natural environments where $S^*$ would be expected in the vicinity of the minimum resource limit for net population growth (Tilman's $R^*$) [38].

Steady-state substrate-limited microbial growth has been routinely modeled as a hyperbolic function of the concentration of a limiting resource either internally to the cell, as is the case for the Droop model [39], or in the external medium, as is the case for the Monod model [40]. Discussion of the well known Monod and Droop models can be found elsewhere (e.g., [41]), but the Blackman model [42] is a somewhat forgotten model of growth. The Blackman model describes steady-state microbial growth kinetics by a linear dependence on a limiting factor, similar in form to (Eq 15), which persists over a discrete interval until another limiting factor is encountered. Despite their differences, critical evaluation of these foundational microbial growth models and their later modifications (e.g., [43, 44]) have shown support for each [45]. Intriguingly, Blackman kinetics, which were presumably known to Monod some 40 years later, would have provided a better fit to his chemostat data from glucose-limited *Escherichia coli* than did his own empirical model. This realization was the motivation behind several subsequent studies which expanded on Blackman kinetics [46, 47]. Piecewise linear growth dependence on substrate concentration is readily observed in natural microbial communities (e.g., Fig 4 in: [41]), though it is rarely attributed as such (c.f., [48]) and has been largely abandoned in favor of the more mathematically convenient hyperbolic models of Monod and Droop. The competition for membrane space in our proposed model provides a possible mechanism to reconcile these two fundamentally different descriptions of microbial growth, one which could conceivably be put to the test.

The distinction between the assumptions that frame our model from those of the two previously mentioned Models A and B on the dependence of transporter abundance on substrate concentration is fairly subtle, but can predict qualitatively different shapes to the growth dependence on substrate availability. As *in vivo* enzyme turnover number predictions improve and become more widely available (e.g., using machine learning algorithms [43]), the requirement for technically challenging and costly quantitative proteomics data may be obviated. This should enable more comprehensive validation of the model and more broad application to steady-state simulations of microbial growth, requiring only that a genome-scale stoichiometric model of metabolism be available, and that the cell size and the maximum growth rate be additionally known.

## Methods

### Determination of transporter $k_{cat}$ by quantitative proteomics and flux balance analysis

$k_{cat}$ values were determined by Eq 4 using flux balance analysis (FBA) quantitation of $v_{max}^G$ and corresponding protein abundances $n^G$ reported in a quantitative proteomics study [15]. A genome-scale stoichiometric model of metabolism for *Escherichia coli* K12 MG1655

(*i*ML1515; [16]) was used to quantify transport fluxes in simulations of each of the cultivation conditions employed by Schmidt et al. [15] in their quantitative proteomics study.

In their study, *Escherichia coli* K12 BW25113 was grown in batch culture on each of 11 sole carbon sources and harvested in logarithmic phase, or in continuous culture on glucose supplied at 4 dilution rates (0.12, 0.20, 0.35, and 0.50 h$^{-1}$). FBA was implemented by allowing for unconstrained transport of only those substrates which were supplied in each defined medium, and the "biomass reaction" was constrained to the experimental growth rate. An L1-norm minimization was implemented to remove loops. Extensive documentation and guides to the implementation of FBA with *i*ML1515 and other stoichiometric models of metabolism within several programming environments is available at opencobra.github.io. FBA was implemented within the Matlab (The Mathworks, Inc.) Toolbox COBRA (Version 3.0; [21]), the Python (Python Software Foundation) package COBRApy [22], and optimizations were performed using Mosek (Version 9; Mosek ApS).

### Determination of transporter capture area

Ligand capture area *A* for each transporter was interpreted to be the area of the transporter membrane domain. Protein sequences for each transporter were used to model quaternary structures using RaptorX [23]. The predicted structures were then rendered and transmembrane domain dimensions were measured using PyMOL (The PyMOL Molecular Graphics System, Version 2.3 Schrodinger, LLC).

### Estimation of molecular diffusivities

Aqueous diffusion coefficients were determined as an empirical function of hydrated molecular volumes and water viscosity [24]. Hydrated molecular volumes were calculated using the LeBas incremental method [25]. Dynamic viscosity was calculated as a function of temperature and salinity [26] of the cultivation conditions [15].

### Physiological data

To their credit, Schmidt and co-authors [15] had the foresight to supply detailed and complementary physiological measurements with their proteomics dataset. Electron micrographs provided measurements of inner and outer cell membrane dimensions, and cell concentrations and growth rates were reported with each medium formulation. We commend the authors on providing these data in an accessible format, as it has enabled our model validation and development. It is worth noting that although a different *Escherichia coli* K12 substrain (BW25113) was used for the majority of growth conditions to quantify protein abundances from the metabolic model substrain (MG1655) we used in FBA simulations, Schmidt et al., [15] showed only minor differences in absolute protein quantitation between these two substrains in glucose-replete batch cultures.

### Supporting information

**S1 Fig. Comparison of the abundances of permease domains of known glucose transport systems in *Escherichia coli* K12 grown in glucose-replete batch cultures and glucose-limited chemostats, as determined by quantitative proteomics ([15]; Data plotted with permission from the authors).** MalF—maltose permease; ManY—mannose permease; MglC—galactose permease; PtsG—glucose permease.
(TIF)

## Author Contributions

**Conceptualization:** John R. Casey, Michael J. Follows.

**Data curation:** John R. Casey.

**Formal analysis:** John R. Casey, Michael J. Follows.

**Funding acquisition:** John R. Casey, Michael J. Follows.

**Investigation:** John R. Casey, Michael J. Follows.

**Methodology:** John R. Casey, Michael J. Follows.

**Software:** John R. Casey.

**Supervision:** Michael J. Follows.

**Validation:** John R. Casey, Michael J. Follows.

**Visualization:** John R. Casey.

**Writing – original draft:** John R. Casey, Michael J. Follows.

**Writing – review & editing:** John R. Casey, Michael J. Follows.

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
