## [Decision Letter · Decision Letter 0]

9 May 2020

Dear Dr. Casey,

Thank you very much for submitting your manuscript "A steady-state model of microbial acclimation to substrate limitation" for consideration at PLOS Computational Biology. As with all papers reviewed by the journal, your manuscript was reviewed by members of the editorial board and by several independent reviewers. The reviewers appreciated the attention to an important topic. Based on the reviews, we are likely to accept this manuscript for publication, providing that you modify the manuscript according to the review recommendations.

Sincerely,

William Cannon

Guest Editor

PLOS Computational Biology

Mark Alber

Deputy Editor

PLOS Computational Biology

[LINK]

Reviewer's Responses to Questions

**Comments to the Authors:**

Reviewer #1: Summary

In this study the authors have proposed a new model to describe the acclimation of cell surface transporter numbers in response to substrate availability in the external media. The authors validate the model using composite mathematical model that combines the stoichiometric model with a kinetic model of substrate transport to predict the cell surface transporter numbers under different substrate concentration. The transporter numbers predicted by the model are similar to the experimentally estimated transporter numbers.

Decision

Minor revision

comments

1. The authors estimated the optimal uptake rate of substrates using FBA. It is unclear how did they achieve it. Did they fix the growth rate and minimize the substrate uptake rate? It is worth explaining this procedure in the text

2. The authors claim that they used the experimental transporter numbers and equation 4 to calculate the Kcat. Is it a single point estimate? How many growth rate vs transporter number data sets did they use to calculate Kcat.

3. In the manuscript the authors compared the model predicted transporter numbers with experimental values. It would be worth comparing the transporter number profile predicted by other models under different substrate concentrations also. This would help understand how different models fair in comparison to the experimental data.

4. Authors assume that under low concentrations the diffusion limits the uptake rate. However, when the external environment is well mixed the rate limiting step will be the encounter rate. Will the proposed model for acclimation hold under this condition?

5. The authors compare their prediction with the chemostat data. How did the authors confirm that the highest dilution rate corresponds to the growth limit? In the chemostat it will be very difficult to reach the growth limit as it would lead to wash out.

Reviewer #2: The authors examine a model that describes a single cell growing on a single limiting substrate and use flux balance models to estimate fluxes from glucose-limited chemostat data and then make comparisons to estimate parameters in a piecewise linear steady-state acclimation model of transport. The authors draw comparisons with the Blackman kinetics model, which has been used for analyzing chemostat data in the literature. As the authors note, recent progress in quantitive proteomics allows analysis of resource allocation. The authors methods are sound and adequately described to allow reproduction.

Minor comments

The authors note in line 154 that for ambient substrate concentrations below the nutrient-limited concentration (which they term the diffusion limited regime), that there is a dependence on the cell radius (i.e., size) and that volumetric changes at different glucose-limited dilution rates. The authors should describe what modifications to the model would need to be made in order to account for the change seen in E. coli and to species with greater observed changes in cell size.

In the Model Validation section (beginning on line 229), the authors describe some of the results from the model and comparisons with literature values, most of which are the same order of magnitude. The authors adequately discuss differences with zinc, which had the greatest difference, but should also discuss the differences observed with acetate.

Reviewer #3: The authors investigate the impact of diffusion and growth on the optimization of the number of transporters for the uptake of a limiting metabolite. The paper is written in a clear language and has a logical flow. The description of the model and justifications of assumptions based on existing data are also clear and accessible. Their use of different sources of data (including proteomics, FBA, and computational chemistry) to inform the model is also commendable. In my opinion, the paper is a valuable contribution and is a nice example of developing a model to explain the physics behind nutrient uptake and its regulation.

Minor comments:

1. It would be nice to have data points for glucose concentrations between 10 uM and 30 mM in Figs 1 and 4. However, I understand that if the authors are relying on existing data, adding these data points may not be feasible.

2. Please consider explicitly mentioning what parameter is being fitted for the model in Fig 4.

3. If I’m interpreting Eq (14) and Fig 5 correctly, the model assumes that at high concentrations of glucose the growth rate does not change. In our empirical observations with E. coli growth (coincidentally an MG1655 strain) at different glucose concentrations, the growth rate remains fairly steady at low and intermediate levels of glucose, but drops at very high concentrations (e.g. a significant drop at 20 mM of glucose). Since a change in the growth rate means a change in the biosynthetic rate (Fig 2), I am curious if such a drop is either predicted from the current model or maybe alternatively (if it is caused by another mechanism) partially explains the lower transporter numbers at high glucose concentrations.

4. Possibly relevant to the previous comment, when comparing with experimental data, it would be helpful (although not necessary) to estimate up to what concentration the assumption of “growth under a single limiting resource” is valid.

**Have all data underlying the figures and results presented in the manuscript been provided?**

Reviewer #1: Yes

Reviewer #2: Yes

Reviewer #3: Yes

PLOS authors have the option to publish the peer review history of their article (what does this mean?). If published, this will include your full peer review and any attached files.

Reviewer #1: No

Reviewer #2: No

Reviewer #3: Yes: Babak Momeni
---

## [Editor Report · Decision Letter 1]

9 Jul 2020

Dear Dr. Casey,

We are pleased to inform you that your manuscript 'A steady-state model of microbial acclimation to substrate limitation' has been provisionally accepted for publication in PLOS Computational Biology.

Best regards,

William Cannon

Guest Editor

PLOS Computational Biology

Mark Alber

Deputy Editor

PLOS Computational Biology

---

## [Editor Report · Acceptance letter]

19 Aug 2020

PCOMPBIOL-D-20-00451R1 

A steady-state model of microbial acclimation to substrate limitation

Dear Dr Casey,

I am pleased to inform you that your manuscript has been formally accepted for publication in PLOS Computational Biology. Your manuscript is now with our production department and you will be notified of the publication date in due course.

With kind regards,

Matt Lyles
